# Disaster Recovery Practices and Resilience Building in Greece

**Harry Coccossis** [1],*[ID] **, Pavlos-Marinos Delladetsimas** [2] **and Xenia Katsigianni** [2][ID]

[1] Department of Planning and Regional Development, University of Thessaly, 383 34 Volos, Greece
[2] Department of Geography, Harokopio University of Athens, 176 71 Athens, Greece;
p.delladetsimas@hua.gr (P.-M.D.); katsigianni@hua.gr (X.K.)
* Correspondence: hkok@uth.gr

**Abstract:** This paper aims to elaborate on the notion of resilience by analysing the historical long-term impact of recovery processes that follow catastrophic events. In this respect, the approach reveals the importance of two major dimensions of disaster recovery practices: the establishment of an effective resilience milieu in conjunction with the generation of safety knowledge. The analysis focuses on two island case studies in Greece that experienced devastating earthquakes in the 1950s: Cephalonia (Ionian Sea) and Santorini (South Aegean Sea). Both insular cases underwent a comprehensive and (in many respects) innovative reconstruction process that set the scene for establishing a 'resilience milieu' and, in a dialectical manner, a 'safety culture', which for many years has been embedded in local development trajectories and influenced spatial growth dynamics.

**Keywords:** resilience building; safety culture; disaster recovery; reconstruction; risk perception

## 1. Introduction

Anticipating and incorporating change has been at the core of spatial policy concerns in recovery-reconstruction processes following disasters, in the sense of looking for new desired spatial structures, amenities and functions to achieve better living conditions and a safe environment. Likewise, spatial policy has been instrumental in resilience building in several cases, by restoring the effects of unanticipated change caused by man-made or natural disasters. In such contexts, it is often necessary to restructure policy processes and implementation practices to overcome time constraints and limitations, as well as eventual resource deficiencies (human, administrative and financial) in coping with exceptional change impacts. In such conditions, the key challenges are time, resources and institutional arrangements, within an overall task to overcome eventual disruptions and catastrophes of the spatial fabric and functions. Resilience building has been at the forefront of policy and governance priorities responding to escalating risks (anticipated or unforeseen) in various contexts, ranging from broader technological and socio-economic changes to climate change. In particular, resilience building is reconsidered, redefined and reshaped after major catastrophic events and on the basis of disaster knowledge acquired. In this context, resilience is a widely endorsed notion in spatial analysis and practice, encompassing multidimensional processes of capacity building in terms of institutional preparedness, improving resistance of physical structures and developing community capabilities to cope, recover and learn from crises and disasters.

While unfolding the different facets of resilience, this paper gives prominence to the way in which disaster events and subsequent recovery policies provide a noteworthy impetus for resilience building. It stresses the dynamics that determine and transform disaster experiences (institutional, social, and technical) stipulating in turn—in a reciprocal manner—the generation of an embedded safety culture. The paper therefore provides new insights into the understudied area of the articulation of recovery planning and resilience building [1,2], in concordance with conceptual approaches that underscore the opportunities arising in the aftermath of disaster events [3–5]. Against this backdrop, the

paper argues that the disaster recovery phase can be seen as a period when major changes take place and within which latent qualities and capacities are reshaped, stipulating new (improved) socio-spatial structures, changing risk perceptions-behaviours and setting the ground for the development of safe communities. This paper focuses also on the distinct risk context of insular settings, in which planning for disaster contingencies cannot involve the adoption of the same terms and criteria as for inland regions. Isolation, transport accessibility and societal features are the most important domains in the study of disaster recovery practices and resilience building [6]. An island context' implies, above all, the need to deal with a variety of spatial insular typologies, and it is predominately defined by two major sets of parameters: (a) the geographical uniqueness—by definition—of the island, arising from its inherent physical and socio-economic characteristics as shaped by the conditions of remoteness, isolation and self-sufficiency [7]; and (b) the exceedingly unpredictable and all-encompassing hazard: the earthquake; accompanied by a multiplicity of secondary hazards such as, landslides, submarine landslides, volcanic eruptions and tsunamis. These sets of parameters also determine the post-disaster response and recovery patterns as well as the type, scale and duration of external support. To address these issues in more depth, two islands of different size and developmental trajectories (Cephalonia-Santorini) serve as case study areas.

The paper is thus structured as follows: Section 2 provides an overview of existing resilience building theoretical and conceptual approaches and presents a conceptual analysis of 'disaster recovery' processes highlighting the role of adopted practices and planning principles that enable and strengthen a resilience milieu. Section 3 elaborates on the concept of safety culture and underlines the potential of disaster knowledge acquired in resilience building. In Section 4, the research approach and methodology of this study is described leading to Section 5 that takes two historical case studies (Cephalonia and Santorini) and analyses their post-disaster recovery-reconstruction processes and the respective building of a resilience milieu and disaster knowledge creation. The following section presents some key findings and the final section concludes with considerations of possibilities for broadening the framework for resilience studies in spatial planning.

## 2. Resilience Building and the Role of Disaster Recovery Process—Literature Review

Resilience traces its origins as a theoretical concept in physics and ecology; resilience was thus used to study resistance to shocks, based on the material's structural and physical characteristics and, respectively, the ability of ecological systems to absorb and adjust to change. Out of these domains, the evolution of the related literature has involved both static perceptions of the concept in line with a techno-scientific rationale—framing the engineering dimension of resilience—and more dynamic ones intertwined with the numerous processes activated when a major change (disaster or shock) occurs. Systems that undergo change actuate mechanisms to confront stresses, recover and return to former conditions of stability and equilibrium. These have been thoroughly unveiled in resilience studies developed in the field of ecology [4,8,9], which in turn constitute the theoretical underpinnings of resilience discourse in disaster social studies. In this vein, several scholars have associated resilience to psychological dimensions related to the analysis of the way people face risks and traumatic experiences and in concordance to withstanding adversity and recovering abilities [10]. In turn, resilience dimensions have been assimilated in disaster management studies, placing emphasis on the psychological processes that people experience when exposed to disaster risks assessed by means of collective and individual vulnerability parameters [11–13]. Overall, resilience provides a robust analytical basis for the study of disasters that allows the combination of a wide range of factors including the quality of physical assets and infrastructure, institutional-governance patterns, critical socioeconomic features, and public perceptions and behaviours. As such, resilience has also been gaining attention in applied policy fields (e.g., Young Foundation 2010, Resilience Alliance 2010, Rockefeller Foundation, UNDRR, etc.) and tends to replace (or complement) notions such as 'adaptivity', 'sustainability' and 'vulnerability' [5], by including dynamic

variables enabling the study of disaster impacts and recovery as an integral socio-spatial continuum and as operational processes.

In principle, resilience refers to the capacity of a community, facing risks, to resist or adapt to eventual impacts and changes efficiently [14]. This capacity is multidimensional and relates both to physical, institutional and socio-economic domains. Resilience is also defined as a place-based and context-specific notion [15,16] built on historical practices and disaster experiences that create opportunities to strengthen local organizational capacities, in support of subsequent socioeconomic and spatial development [1]. To this end, these local organizational capacities, as derived from the nexus between disaster recovery planning and resilience, stipulate the concept of 'safety culture', which is inherently linked to response experiences and knowledge acquired. As a matter of fact, among the elements that enable resilience building, several scholars [16–18] underline the importance of issues such as: the quality of the pre-disaster physical fabric (e.g., age and status of buildings and infrastructures), emergency and post disaster-recovery experiences, the efficacy-quality of repair and new building construction [19–21], the institutional/governance capacity for disaster management, reconstruction [22,23], infrastructure and spatial planning [24,25], and risk perception [26]. Among a variety of conceptual and methodological frameworks that have been developed, Manyena [27] identifies two major approaches of resilience. The first focuses on the quality of physical structures and institutional responses based on command-and-control strategies, emergency services and operational plans, while the second underscores the social dimension of resilience relying on dynamic processes, through which communities acquire certain capabilities to face disasters. Lately, place-based models gain attention as they manage to provide integrated frameworks combined with quantifiable components for assessing context-specific resilience patterns [15,16]. The common ground of all these approaches is that resilience mostly relates to three main capacity building facets [28] including: coping capacity, adaptive capacity and transformative capacity. Moreover, existing frameworks bring to the forefront two distinct resilience components [16]: one, intrinsic and the other, adaptive. The first refers to an effective and safe functioning of communities in normal development periods, while the second refers to the ability-flexibility in responding and recovering after disaster situations. In the context of this paper, it is argued that the two elements are intertwined, given that the ability to recover and the processes of recovery per se often shape and influence subsequent resilience qualities characterizing communities in normal operations.

From a spatial planning perspective, disaster recovery planning sets the ground for the delineation of various resilience dimensions. Recovery deals with a number of activities that include the provision of temporary housing, the reconstruction of damaged assets (buildings and infrastructure), the development of new settlements or residential zones, the restoration of public services, and the reestablishment of socio-economic development. These actions permit, and authorize, a conditional intervention into long-lasting institutional structures and governance systems in order to facilitate rapid processes and accelerate the return to normality, offering the opportunity to detect malfunctions and causes of breakdowns. Similar to the way reconstruction results in reinforced built fabric—and thus resistant to subsequent catastrophic events—recovery planning can trigger the establishment of new institutions (local or regional), the improvement of vertical and horizontal coordination in disaster management, as well as the enforcement of updated regulatory frameworks (e.g., enhanced building codes and risk mitigation measures, urban plan amendments incorporating safety elements). In this context, recovery creates an 'added value' that influences governance structures and the development of a knowledge base that is absorbed at various levels of disaster management decision-making [29,30]. Resilience characteristics are built upon the aforementioned processes and 'learning from errors' in a constant attempt to reorganize material and immaterial components of a spatial system [25] and improve its capacity to deal with unexpected events and crises [3]. Disaster and recovery experiences have (in many cases) activated learning mechanisms leaving a legacy of knowledge, which in turn has demarcated subsequent development trajectories.

This is effectuated through influencing social perceptions and attitudes, transforming social learning in new institutional arrangements, regulations and spatial strategies. In this respect, a resilience becomes what Manyena [27] describes as the capacity of a place to 'bounce forward' and move on following a disaster; here, 'capacity' is defined as the inherent mobilization of physical, institutional and socio-economic factors as well as human resources and expertise. All these imply that resilience is a learning process driven by various actors [31] and shaped by societal choices (normative and informal) 'based on principles derived from past disaster experiences' [32] (p. 2). Thus, resilience building in spatial systems should be seen as a dynamically evolving development between historical catastrophic events and capacity building.

### 3. Disaster Experience and Safety Culture

Apart from spatial, physical and institutional capacities that set a local community resilience milieu, the impact of catastrophic events is also dependent on the dominant social behaviour and risk perceptions as dialectically shaped within the prevailing cultural norms and beliefs—a principle described as *safety culture*. The term emanates from high risk and uncertainty milieus and the search for an appropriate conceptualization of risk management that bring behavioural parameters into the analysis [33]. This analysis appeared prevalently in the aftermath of the 1986 Chernobyl disaster [34] and was developed as a concept complementary to organizational culture, defining the ability of an organization to manage risks and achieve safe operation at various levels [35]. Safety culture is formulated at national and regional level through the enforcement of regulatory frameworks, wider systems of rules, as well as by differentiated factors such as the use of technologies, the operation of several organizations and the strategic visions of their leaders [36]. As Cooper ([37], p. 113) underlines, safety culture '*does not operate in a vacuum: it affects, and in turn is affected by, other non-safety-related operational processes and organizational systems*'. Although the broader notion of 'culture' implies the existence of wider systems of beliefs and norms that influence perceptions and guide social action, the analysis of safety culture is rather confined to individuals and group attitudes associated with risk taking behaviours and the conformity to safety regulations [38]. However, Turner [39] stresses the need to approach safety culture as a sociotechnical feature and not merely a social construct—a perspective that further validates its relevance to disaster management and resilience literature.

For Geertz [40] (p. 145), safety culture is a conceptual fabric through which human beings interpret disaster experiences, assimilate knowledge and adapt their actions to reducing (individual and collective) vulnerability. In this respect, past disaster experiences assume an accentuated role since they enable understanding and building response capacity to cope with disasters. It is the added value of new knowledge acquired during recovery processes (described in the previous section) that gradually alters dominant perceptions and attitudes, reciprocally transforming resilience building into an embedded safety culture. Set in this context, safety culture and risk perception overlap, involving factors such as beliefs, attitudes, social practices and values [41]. Kasperson et al. [42,43] rigorously stress that a catastrophic event activates a series of social, psychological, and cultural processes through which risks become recognized, amplified and/or attenuated. In a similar context, Renn [44], in his thesis on the social amplification of risk, posits that events pertaining to hazards interact with psychological, social, institutional and cultural processes in ways that can heighten or attenuate public perceptions of risk and shape risk behaviour. Behavioural patterns, in turn, generate secondary social or economic consequences, which extend far beyond direct disaster losses stimulating significant indirect effects. Moreover, disaster experience strengthens risk perception and awareness [45–47], increases the individual responsibility for preparedness and motivates communities to adopt risk mitigation measures [48]. However, many communities, in the long run, have often tended to marginalize acquired safety knowledge [49,50], bouncing back to "unsafe" practices, overwhelmed by the dominant growth (or decline) dynamics. Several scholars underline the importance of solidarity mechanisms [51,52] that emerge in times of crisis,

interweave in local or international networks and manage to ensure a duration in terms of knowledge transfer that, in turn, strengthens disaster memory. Paidakaki and Moulaert [53] even argue that a "resilient city cannot exist per se, but rather social resilience cells (SRC) within the city unfolding their own transformative capacities and up scaling experiences." All in all, multi-level knowledge acquired after a disaster becomes part and/or influences attitudes and local mentalities and evidently risk perception. Behavioural 'social arenas' are consolidated through decision making processes operating under conditions of structured rules of interaction' [43]. What is more, the integration of local knowledge in risk mitigation measures leads to more effective risk mitigation strategies and thus, strengthens resilience building [54].

## 4. Material and Methodological Considerations

This study draws on theoretical insights that conceptualize resilience as a complex socio-spatial transformative process and focuses on the critical role of disaster recovery processes that (re-)activate resilience building at a physical, institutional and community level. In this context, we bring into the analysis the concept of 'safety culture' in order to examine the impact of recovery practices and implemented policies in the long run, based on the assumption that they create an added value to local communities. This added value refers to various forms of experiential and transmitted knowledge [55], which—albeit underestimated in disaster studies [56]—influences perceptions, attitudes as well as governance structures and decision-making while generating a 'new' resilience milieu. Therefore, a place-based approach of resilience is adopted in this study allowing a thorough examination of historical disaster events and subsequent recovery policies in two distinct case studies in Greece: the islands of Cephalonia (Ionian Sea) and Santorini (South Aegean Sea). The rationale for selecting these insular areas lies on the fact that they both experienced catastrophic events in the same historical period, allowing for structuring a comparative framework as they shared common (national) socio-political and institutional dynamics. In addition, both cases embarked in radical reconstruction processes, stipulating virtually new spatial structures, also involving the attempt to introduce safety elements in the post-disaster era of the affected areas.

The first step of the study is thus to examine and critically analyse the making of a new resilience milieu during disaster recovery period, in terms of new physical and urban structures produced, institutions set up to facilitate post-disaster reconstruction and spatial plans developed to meet the needs for safe development trajectories. In this respect, research methods included desktop research, extensive review of literature and of studies that incorporated primary data of disaster experience and recovery practices for the two cases), and document analysis related to emergency management guidelines, sheltering and housing provision, recovery policies, urban plans and building codes developed or amended amid and after the disaster recovery period. We further explored sources and documents that refer to subsequent risks and catastrophic events taking place on the islands, in order to examine to what extent the local communities effectively "confronted" subsequent disaster risks. By mobilizing the concept of safety culture, we further reflect on how catalyzing disaster and recovery experiences produced a long-term resilience milieu, which was also structured by different (and adverse) post disaster growth trends in the two communities. In light of these considerations, we reconceptualize the resilience embedding process in conjunction to the added value of disaster-lessons learned.

## 5. Case Study Analysis

This section uses the theoretical context of resilience described above to provide a retrospective analysis of islands facing disastrous conditions as a consequence of earthquake destructions. The study focuses on the experiences of the islands of Cephalonia and Santorini and their related recovery processes (Figure 1). In the aftermath of World War II, these two insular communities experienced devastating earthquake disasters in 1953 and 1956 respectively (forming part of a catastrophic earthquakes series that affected several

areas across the Greek territory: the Ionian Islands <Cephalonia-Zante-Ithaka> in 1953, Volos agglomeration; in 1954 and 1955, Amorgos and Santorini in 1956, and Rhodes in 1957). Comparing the two cases reveals the extent to which the 1950's disasters influenced resilience building and generated a safety culture in the local communities. Notwithstanding the comparison drawn regarding the post-disaster recovery experiences, the islands exhibit distinct characteristics such as their demographic composition (population size, age strictures and population variations between summer and winter periods), their settlement structure and building stock quality, their economic accessibility/transport services in combination with the (pre-existing) local resilience potential.

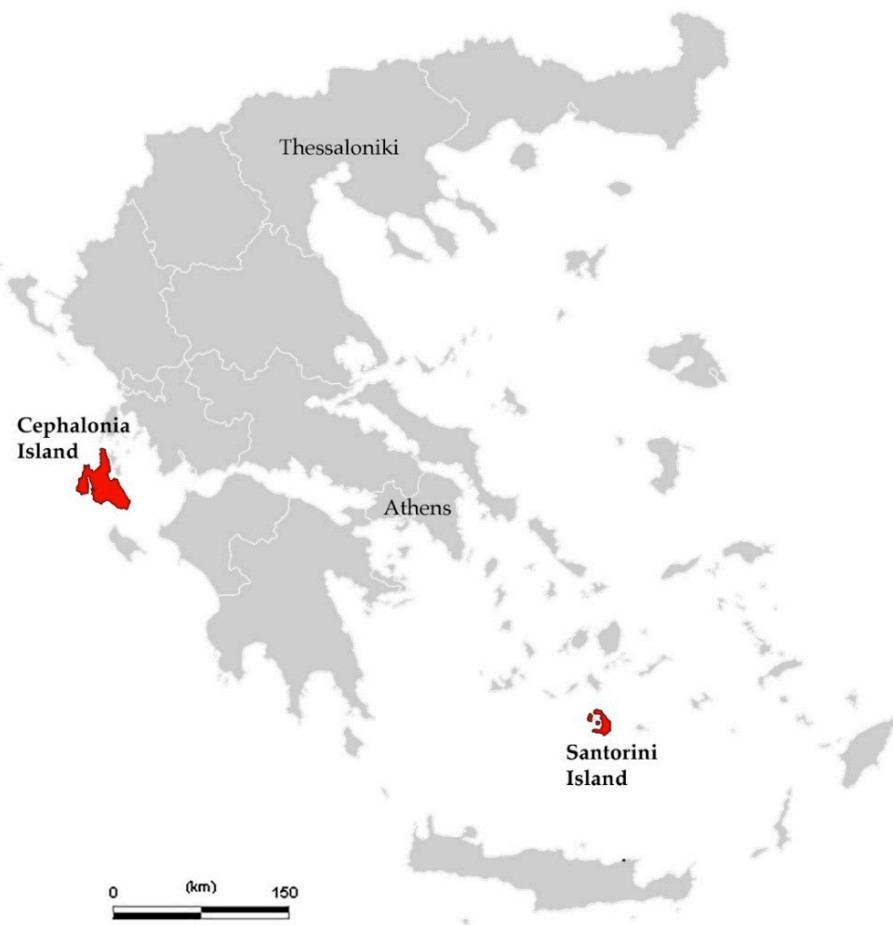

**Figure 1.** Location of case study areas.

*5.1. The Case of Cephalonia: The 1953 Disaster and Recovery*

In the case of Cephalonia (Ionian Islands Periphery-Western Greece), the recovery process from the 1953 earthquake is considered a unique recovery milestone for Greek policy as a whole. The recovery process had an outmost long-term catalysing impact not only at a local but also at the national level, defining the entire post-war policy in terms of adopted recovery-reconstruction practices and measures. The island is characterized as the most seismically vulnerable zone in Europe, defined as a fringe area connecting different geodynamic potentials. The island has historically experienced consecutive catastrophic earthquakes resulting in excessively high magnitude disastrous events. Emphasis here is given here to the devastating earthquake that occurred between 9 and 12 August 1953, involving a series of events with magnitudes ranging from 5.1 to 6.8 Richter (9 August, <M 6.4 >, 11 August <M 6.8>, 12 August <M 7.2>) causing 471 human losses, 2412 injuries, and massive destruction of the built environment on the islands of Cephalonia, Zante and Ithaka. Of a total of 33,300 housing units, 27,659 collapsed; the entire social and

technical infrastructure of the islands was devastated; architectural and cultural assets suffered irreversible losses. The economic loss reached 200,000,000 US$, which amounted to 3–6% of the country's GDP [57]. The 1953 earthquake also marked a critical brake in the developmental trajectories of the island, causing dramatic population outmigration flows, changes in the demographic composition and a radical restructuring of the physical, social and economic fabric of the affected areas. Set in this context, the recovery reconstruction process was founded on the making of a resilience milieu and a safety culture. The aim of this recovery process was to transform the disaster experience by fostering resilience building, social learning, and the development of a safety culture—in order to further strengthen risk perception and pro-active action.

5.1.1. The Making of a Resilience Milieu

The making of a resilience milieu has consisted of mainly two interlocking components: the implementation of building-physical planning measures and the exceptional introduction of institutional structures.

Building and Physical Planning Measures

In regard to the building-physical planning components, the process led to the introduction of a rational comprehensive planning recovery process involving: (i) The provision of systematic emergency housing including tents, provisional shelter, and permanent housing (Figure 2); (ii) An exceptional investment process in infrastructure provision (repair and new construction) comprising the road network, new ports infrastructure, energy and sewerage networks; (iii) The implementation of new land-use plans covering all towns and villages of the islands leading to the realization of a new radical spatial structure replicating (locally) the dominant "modernist" planning ideas of the period; (iv) An extended relocation-concentration of rural settlements and creation of new towns accompanied by a re-parcelling and property systematization processes in land policy; (v) The development of a highly innovative permanent housing solution including a 'self-help nucleus' or 'site and services' via the use of household labour and the central state as well as the free distribution of building materials, building unit design plans, building construction supervision and development control; (vi) The enactment of the first Seismic Safety Code (1959) enforced at the national level, forming the basis of the entire post war earthquake safety policy.

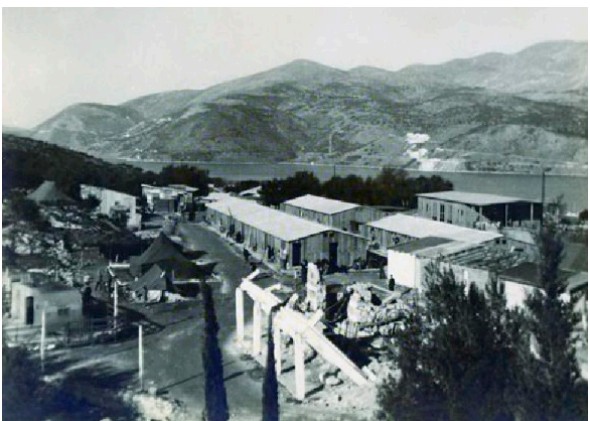 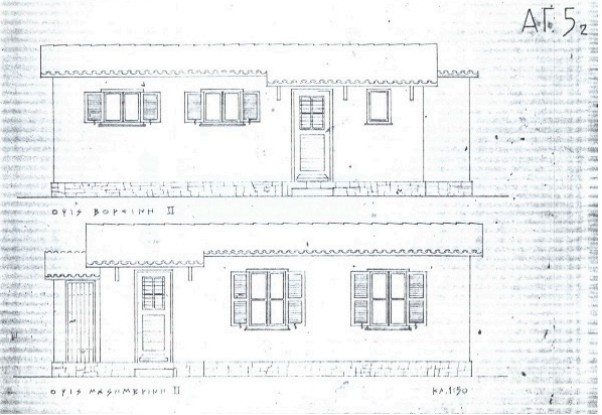

**Figure 2.** Provisional shelters in Cephalonia-Argostoli (**right**). Typical site and services housing (**left**) typology [57].

The first national Seismic Code was enacted in 1959 after the Ionian Islands earthquake and was based on decrees extraordinarily introduced for the Ionian areas. The seismic code experienced consecutive improvements (in 1985, in 2000 and supplemented in 2003). The seismic code lies at the heart of the increased resilience and building structural performance of Greek building assets (and obviously Cephalonia itself) following earthquake events throughout the entire post-war period.

Institutional Components

The institutional resilience components that were put forward with the Ionian recovery consisted of: (i) The enactment of the "Sub-Ministry of Reconstruction" institution, which accumulated all recovery-planning competences for integrally managing and accelerating the recovery-reconstruction process. This institution became not only the major recovery entity but also an exclusive planning institution comprising in its statute the entire spectrum of competences such as physical planning, development control, social and technical infrastructure, welfare and economic policy; (ii) The creation of new entities managing and systematizing, on an extended basis, mortgages, compensations and rebuilding-repair grants.

### 5.1.2. Safety Culture

Overall, the 1953 experience contributed to numerous resilience components that allowed the insular community to confront subsequent disaster events occurring throughout the post-war period. More specifically, since 1953 the island has constantly experienced consecutive earthquakes and other natural and technological disasters. The most important of these being: (i) the 1983 earthquake (M 6.7), (ii) the 2014 earthquakes (M 6.1 and M 6.0), (iii) successive forest fires (highly disastrous especially in 2001, 2007 and 2011), (iv) extreme weather conditions resulting in a major power failure in 2010, (v) numerous landslides throughout the years and (vi) the 2020 hurricane disaster. All these, in combination, clearly delineate a multi-hazard hazard risk environment reflecting at the same time an embedded safety culture, taking into account the positive response performance of the local institutions and population [58]. Of particular relevance to highlight all the above is the February 2014 earthquake disaster that affected the island, following two consecutive geo-dynamic episodes (on 26 January and 3 February 2014; 26/01/2014 ML = 5.4, PGA = 0.58 g 03/02/2014 ML = 5.7, PGA = 0.68 g; <epicentre area> ML = 5.7, PGA = 0.77 g). The seismic events did not cause human losses (only minor injuries); the earthquake was accompanied, however, by indirect cascade effects (landslides, soil liquefactions and raptures) that in combination have seriously affected housing, commercial assets, the road network, public utility networks and port infrastructures. The earthquake was indeed an all-embracing disaster causing additional secondary effects to the economic and social structure of the insular setting [57]. However, taking into consideration the magnitude and especially the astonishingly high acceleration levels of the seism, the structural efficacy of the buildings has been exceedingly positive. All in all, it has reflected the long-standing safety culture [58] embodied in the island as a remnant of the 1953 experience. The island underwent an overwhelmingly proficient emergency mobilization (first aid- search and rescue-food provisions and emergency supplies, emergency road network management and signalling, emergency shelters provisions, exceptional financial support to households and businesses, damage-loss registrations and assessments) under the shared responsibility of central state and local-peripheral institutions. In addition, the island has been characterized by advanced behavioural response patterns of the local population and the swift participation of numerous local NGOs and social groups. It could be argued that the legacy of these can be traced to the 1953 experience relating to the locally embedded knowledge accumulated in the implementation of the safety codes in construction (local engineering offices, construction workers, innovative building materials etc.) and the growing awareness of the local communities, as this has been dialectically shaped in combination with the institutional-legislative paths at the national-regional level.

### 5.2. The Case of Santorini: The 1956 Disaster and Recovery

In the case of Santorini, the earthquake of 9 July 1956 was a devastating event that affected the physical fabric, the socio-economic environment (causing major population outflows and a fierce socio-economic crisis for the local community) and the long-term developmental trajectories of the island up to the early 1970s. The earthquake (M 7.5 and 45″ duration) was caused by a seabed fault activation. The catastrophic effects were further aggravated by a subsequent M 6.9 aftershock, and chain hazard events, such as: tsunami waves (reaching 25 m on the southwest coastal area) and landslides that were perpetuated for several days. Fifty-three (53) human deaths were recorded and more than 100 injuries, while massive damages were caused to the island's building stock. Of a total of 4000 buildings 1041 collapsed and 1402 required repair. Moreover, the earthquake caused massive damage to the road network, which consisted primarily of one principal arterial road, as well as public buildings and services. During the recovery period, several areas—especially on the Caldera slopes (Figure 3)—were evacuated and left abandoned due to soil instability. In Santorini, the recovery-reconstruction period officially lasted until 1963 and as in Cephalonia, involved the implementation of a number of innovative policy elements, which to a certain extent contributed to resilience building for a rather limited time-span.

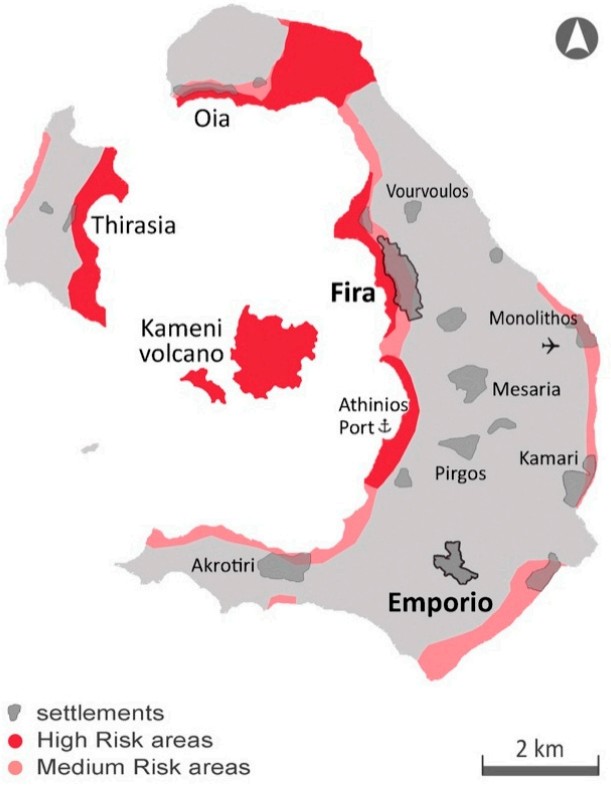

**Figure 3.** Risk zones in Santorini based on volcanic, seismic and landslide risk assessments [59,60].

### 5.2.1. The Making of a Resilience Milieu
Building and Physical Planning Measures

Among the greatest achievements of Santorini's reconstruction was the fact that it was conducted in the context of an excessively remote environment lacking fundamental technical and social infrastructure. Among the objectives of the recovery plan was an accelerated process of housing provision preserving the built heritage and architectural characteristics while at the same time introducing changes in building densities, the transportation network and public service locations [61]. Furthermore, due to the conditions of remoteness and rather inexistent transport infrastructure at the time (absence of port

and airport), building materials and equipment could not be imported to cover the needs of this large-scale reconstruction process. To address this issue, as well as the deficiencies of skilled labour, an innovative "building system" was introduced to produce materials locally, making use of the prominent attributes of the insular soil (volcanic rocks, pozzolanic sand and lime). Building materials were thus designed and produced locally and utilized for the construction of standardized (vaulted structure) housing units. These structures proved to be highly resistant earthquake proof constructs, relying mostly on the use of special building tiles (*pumice concrete blocks*) that were locally produced by small-scale installations purposively set up on the island.

Apart from the structural elements that in many respects fostered a resilience building process in Santorini, of particular importance were the adopted land policy measures and planning mechanisms implemented during the recovery-reconstruction process. Namely: (i) The implementation of an overall land systematization and re-parcelling process, also allocating land plots to homeless strata; (ii) A self-help housing system implemented in response to the acute housing needs. In this context, the state provided the beneficiaries with (free) standard architectural design and building supervision, as well as building materials. In the context of this recovery policy, exceptional compensation grants were also provided for repairing or building new constructions. It also involved the construction of an entirely new settlement (Figure 4). (iii) Additional financial support was granted to extended households and to land owners of plots "exhibiting a touristic potential development", introducing an inherent component of post-reconstruction development.

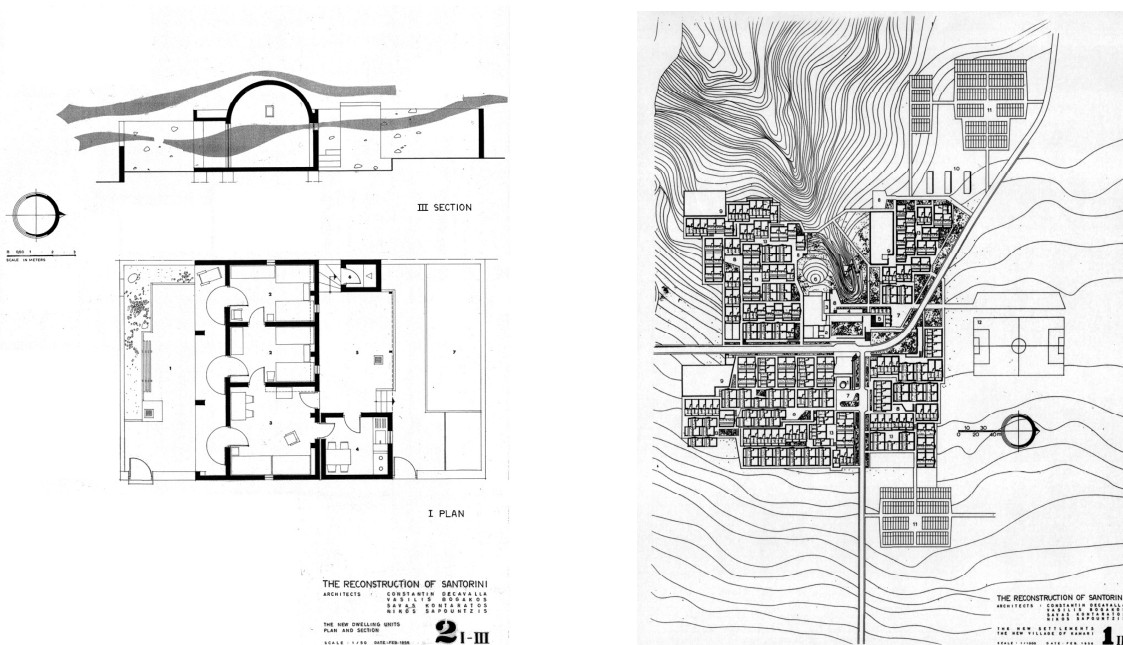

**Figure 4.** The new settlement of Kamari (**right**) and the standardized housing unit (**left**) designed for the reconstruction of Santorini [57].

Institutional Components

To implement the aforementioned, a distinct institution was formulated: the 'Thera Settlement Office.' This office was tasked with the integral coordination of the recovery-reconstruction process of the insular community. In addition, an Architectural Committee was appointed by the Ministry of Public Works aiming at facilitating and accelerating plan approvals and development control procedures. The Architectural Committee introduced a notion of compact development strategy, combining the new redevelopment structures with existing built-up urban fabric, strengthening social bonding within the local community and among neighbourhoods, aiming to reduce population outflow.

### 5.2.2. Safety Culture

It could be argued that the island's reconstruction was a 'bounce forward' for the local community, providing a socio-economic boost based on an accentuated safety culture. Especially when it comes to spatial development planning, the legacy of the past and the knowledge acquired after the 1956 earthquake was transmitted and valorized (manifested partly in the first 1987 Land Use Plan of the island), in numerous attempts to assimilate disaster experiences and produce an integrated preparedness framework and safe development trajectories. For a long period, disaster experience strengthened community spirit and intensified public perceptions of risk that survive up to date, as the disaster is an intrinsic part of the island's history (volcanic eruptions 1925–1928, 1939–1941 and 1950, 1956 earthquake, landslides in 2011 causing 1 human loss and 2 injuries). However, these past experiences have not been transformed into a social learning process [62]. Although Santorini's residents are aware of their exposure to high (volcanic-earthquake) risk as well as of their vulnerability status [56], dominant patterns of spatial and economic development intertwined with the emergence of mass tourism have been operating against resilience building and systematically feeding into a regime of acute spatial and social-cultural vulnerability. Indicatively, built areas on Caldera slopes, which were designated as utterly unsafe after the 1956 earthquake, today constitute some of the most attractive areas in terms of population flows, activity concentration and tourism-related uses. Similar to Cephalonia, Santorini is a multi-hazard environment, shaped by its volcanic and seismic nature (accompanied by severe landslides and manmade accidents), with an exceptional disaster recovery experience that it has not managed to assimilate in a local safety culture. The island's emergency planning provisions (lately formulated) remain detached from spatial planning policies. In addition, the local community does not seem to effectively comply with protection measures at a personal or societal level. The recent activation of the Kameni volcanic centre (2010–2012), which caused around 10 low magnitude seismic events per day in 2011 [63], became the focus of national and international institutions, but had no concrete outcome in fostering renewed reduction and preparedness provisions in the island. Furthermore, the severe landslides that took place in 2011 were managed only with provisional and fragmented actions on behalf of the local authority. There is thus a dominant local 'mentality' of regulating and provisionally overcoming risk events, reflecting an overwhelmingly limited risk perception and failing to generate an embedded safety culture.

### 6. Results and Discussion

The case studies demonstrate both similarities as well as differences in their levels of resilience building. Both islands demonstrate an improvement in their capacity to maximize their ability to receive and distribute external aid and to manage population flows and resources. In both cases, at the institutional-legislative level, the seismic disasters and subsequent recovery policies have, by and large, led to the introduction of new laws, regulations and safety codes (prevalently the seismic code) the value of which has exceeded the specific geographical boundaries and historical time spans. At the preliminary stage, emergency legislative actions had an "area-specific outlook" but later assumed a state-national level importance. They became part of a wider state policy pattern to deal with disaster experiences and have been widely (replicated) implemented in other disaster-affected areas in the country. In relation to the specific insular localities, the implementation of these discrete recovery policies involved the enforcement of new innovative by-laws, spatial plans, building self-help housing schemes and safety regulations, that marked a break with the past practices and determined a new resilience milieu.

The main issue of concern arising from the analysis of these two historical case studies regards the extent to which policy responses and tools introduced in the context of the recovery strategies have been effective in generating a new 'resilience milieu' as a local social construct, which has reciprocally shaped a distinct safety culture. The two recovery experiences reveal that disaster conditions and exceptional recovery demands

caused by earthquakes stipulated, among others, a transformation of state policy, both nationally and locally. This transformation engaged an extraordinary institutional action, the transfer of economic, human and technical resources to the affected localities and to the establishment of political power in ways beyond policy patterns and priorities prevailing during the previous "normal" development periods. These demands had an urgent-time limited character and embodied a potential for exerting structural influence on socio-economic systems and developmental trends. In addition, they promoted changes of pre-existing institutional-organizational structures by involving the joint mobilization of state agencies of various tiers, competencies (planning, self-help housing, land policy, taxation and welfare policy) and by generating new relationships between the state and local societies [64,65].

The resilience milieu developed in the two islands departed from the typical institutional and political models (which had heavily depended on central level initiatives and interventions at the state level and central administration); the actual outcome in terms of effectiveness and duration has been exceedingly diverse. A lot has depended on issues such as: the scale (size) of the island, its geographic features (isolation, remoteness, accessibility) and local capacity (institutional structure and human resources but also community culture and disaster experiences). Cephalonia is a far larger island than Santorini, richer in key resources and with a long-standing growth experience; it has managed to effectively absorb practices and develop a resilience milieu, exhibiting a relative effectiveness when dealing with subsequent disaster events throughout the post-war era. On the other hand, Santorini is a remote small island that had already faced the threat of abandonment in the 1950s, has weak administrative capacities, and scarcity in key resources; it developed a resilience milieu with a limited duration, being overwhelmed by adverse developmental trajectories.

These aforementioned divergences in resilience making are reflected in different patterns of risk perception and safety culture embedded in the local communities. The two cases, in terms of risk perception and safety, relied mostly on developing capacities for proactive actions and for sustaining spatial systems functions. Resilience-making is thus coupled by a learning mechanism through regulatory responses, institutional arrangements, ensuring delivery of support systems to localities. However, when this learning mechanism is not transformed into 'tangible' institutional reforms, the knowledge acquired may gradually fade away and resilience building can be overturned. Two respective studies [58,62] have revealed that safety culture can be seen as an inbuilt characteristic of the local community in the island of Cephalonia. In the case of Santorini, while it is still present –as a remnant of the 1956 disaster- it assumes a rather tacit outlook. The local community embodies a rather informal and disjointed risk awareness (especially volcanic) behaviour, and at the same time appears to be unwilling to make any sacrifices to increase safety, in order to preserve the current income and rent yielding activities. Thus, if risk perception is not maintained (or the local governance system does not embark in any systematic initiatives in information and education to safeguard it), it constantly decreases over time (following the disaster and recovery period). Limited risk perception in turn affects the building of a consistent safety culture. The latter remaining a simple instrumental consideration and detached from embedded social behaviour.

## 7. Conclusions and Future Research Prospects

Existing approaches and perspectives on resilience essentially stress efficiency for response and action, assuming institutional and community capacities to cope with risks and disasters. This is understandable in major spatial contexts where there are structures, mechanisms and experiences in policy planning and programming, and effective implementing actions. In smaller settlements or rural, mountain or island areas there is often a lack of such a capacity, depending to a large extent on external support. It is in such contexts that resilience is highly affected by social attitudes and perceptions [66], while the cultivation of a safety culture especially through exemplar risk management and disaster recovery practices that integrate local knowledge may appear as a solution to enhance

resilience building. However, further research is needed to assess whether unanticipated events strengthen the aforementioned capacities and whether a learning process is established affecting the potential for efficiency and effectiveness in future responses. An interesting inquiry would be to assess the spatial context structure and dynamics affecting risk perception and response potential, examining island size, island complexes and mainland proximity features (i.e., metropolitan, small city or rural area, etc.). In similar terms, special spatial particularities (i.e., environmental, economic etc) affecting development and prospects might be interesting factors affecting risk response features [67]. Another related understudied area involves assessing other eventual constraints in small destinations, such as in tourist areas for example, where there could be a strong seasonality not only in terms of visitors but also in terms of community size and structures, and the interrelated issues of building a resilience milieu and "safety culture". Finally, this study prompts more exploration on whether in larger agglomerations 'a cultural disaster memory' [68] initiative could contribute to the making of a broader culture, incentivizing proactive risk considerations, and increasing the information basis.

**Author Contributions:** Conceptualization, P.-M.D., and X.K.; methodology, H.C.; document and data analysis, P.-M.D., and X.K.; writing—original draft preparation, H.C., P.-M.D. and X.K.; writing—review and editing, X.K., and H.C. All authors have read and agreed to the published version of the manuscript.

**Funding:** This research received no external funding.

**Institutional Review Board Statement:** Not applicable. The study does not involve humans or animals.

**Informed Consent Statement:** Not applicable.

**Data Availability Statement:** Data available upon request from the corresponding author.

**Conflicts of Interest:** The authors declare no conflict of interest.

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
