# Peer review of "Disaster Recovery Practices and Resilience Building in Greece"

_urbansci, doi:10.3390/urbansci5010028_

Round 1

Reviewer 1 Report

The authors of this article present an interest research in disaster experiences. This paper could be interesting for the URBAN SCIENCE readers, but I think a long way off from being fit for publication. I strongly recommend to further investigate the literature, adding more references and to amplify all sections. In particular, I strongly recommend to organise the initial part of this article following the traditional format of the international journals such as Sustainability: "introduction" and "literature review". I strongly recommend to amplify the initial part of this paper with the connections between resilience and disaster recovery.

First of all, the title should be more incisive; the abstract should emphasize the originality of the research.

The introduction should also be structured better, the authors have to indicate how they intend to develop the article. The conclusions are not present and must summarize the research that has been conducted, the authors should also emphasize the future development of the research.

I think that for these reasons the article cannot be accepted in this form

Reviewer 2 Report

The paper does not have these sections: conclusions, nor discussion, nor a state of the art in the international context.

The introduction should be updated with international references. Discussion is missing and the findings can be used for it. Conclusions are missing.

Round 2

Reviewer 1 Report

The authors made some improvements, but not enough to be able to publish the article as it is. The paper does not present the classic subdivision of international journals. I suggest organizing the paper following the classic format.
I strongly recommend to further investigate the literature, adding more updated references and to organize the paper followng the traditional structure: Introduction, Methodology, Results, Discussions and Conclusions (as I suggested in the previous review).

Reviewer 2 Report

The paper has improved with these changes 

Round 3

Reviewer 1 Report

the authors have significantly improved the paper, I think they just need to do a little reflection in relation to the islands. In fact, some critical issues are much stronger if seen in the context of island contexts, the more the case studies are small islands. In this regard, I suggest authors read and quote the following papers:

  1.  Calado, H., Fonseca, C., Vergílio, M., Costa, A., Moniz, F., Gil, A., & Dias, J. A. (2014). Small islands conservation and protected areas. Revista de Gestão Costeira Integrada-Journal of Integrated Coastal Zone Management14(2), 167-174.
  2. Garau, C., Desogus, G., & Stratigea, A. (2020). Territorial cohesion in insular contexts: assessing external attractiveness and internal strength of major Mediterranean islands. European Planning Studies, 1-20.
  3. Mawyer, A., & Jacka, J. K. (2018). Sovereignty, conservation and island ecological futures. Environmental Conservation45(3), 238-251.

Author Response

Dear reviewer,

Thank you very much for your suggestions.

We have raised the issues suggested on island specificities in the Introduction lines 52-66 and in the end (Conclusions) lines 561-566

Thank you also for the suggested references